# Assessment of Characteristics of Imaging Biomarkers for Quantifying Anterior Cingulate Cortex Changes: A Twin Study of Middle- to Advanced-Aged Populations in East Asia

**DOI:** 10.3390/medicina58121855

**Published:** 2022-12-16

**Authors:** Hiroto Takahashi, Yoshiyuki Watanabe, Tomoki Hirakawa, Hisashi Tanaka, Noriyuki Tomiyama, Yuta Koto, Norio Sakai

**Affiliations:** 1Osaka Twin Research Group, Osaka University Graduate School of Medicine, 1-7 Yamadaoka, Suita 565-0871, Osaka, Japan; 2Department of Medical Physics and Engineering, Division of Health Sciences, Osaka University Graduate School of Medicine, 1-7 Yamadaoka, Suita 565-0871, Osaka, Japan; 3Department of Radiology, Shiga University of Medical Science, Seta Tsukinowa-cho, Otsu 520-2192, Shiga, Japan; 4Department of Diagnostic and Interventional Radiology, Osaka University Graduate School of Medicine, 2-2 Yamadaoka, Suita 565-0871, Osaka, Japan; 5Child Healthcare and Genetic Science Laboratory, Division of Health Sciences, Osaka University Graduate School of Medicine, 1-7 Yamadaoka, Suita 565-0871, Osaka, Japan

**Keywords:** anterior cingulate cortex, cortical morphometry, East Asia

## Abstract

*Background and Objectives*: Our aim was to assess genetic and environmental effects on surface morphological parameters for quantifying anterior cingulate cortex (ACC) changes in middle- to advanced-age East Asians using twin analysis. *Materials and Methods*: Normal twins over 39 years old comprising 37 monozygotic pairs and 17 dizygotic pairs underwent 3-dimensional (3D) T1-weighted imaging of the brain at 3T. Freesurfer-derived ACC parameters including thickness, standard deviation of thickness (STDthickness), volume, surface area, and sulcal morphological parameters (folding, mean, and Gaussian curvatures) were calculated from 3D T1-weighted volume images. Twin analysis with a model involving phenotype variance components of additive genetic effects (A), common environmental effects (C), and unique environmental effects (E) was performed to assess the magnitude of each genetic and environmental influence on parameters. *Results*: Most parameters fit best with an AE model. Both thickness (A: left 0.73/right 0.71) and surface area (A: left 0.63/right 0.71) were highly heritable. STDthickness was low to moderately heritable (A: left 0.48/right 0.29). Volume was moderately heritable (A: left 0.37). Folding was low to moderately heritable (A: left 0.44/right 0.28). Mean curvature (A: left 0.37/right 0.65) and Gaussian curvature (A: right 0.79) were moderately to highly heritable. Right volume and left Gaussian curvature fit best with a CE model, indicating a relatively weak contribution of genetic factors to these parameters. *Conclusions*: When assessing ACC changes in middle- to advanced-age East Asians, one must keep in mind that thickness and surface area appear to be strongly affected by genetic factors, whereas sulcal morphological parameters tend to involve environmental factors.

## 1. Introduction

Brain cortical development is regulated by the interaction of intrinsic (genetic) and extrinsic (acquired environmental) influences. Highly regionalized expression patterns of genes in the cortical structure of the brain were reported in a previous study [1], indicating the importance of understanding the characteristics of the regional cortical structure as an imaging biomarker for the evaluation of neurological diseases.

The anterior cingulate cortex (ACC) is considered to be a nexus point where emotion, cognition, and motivational drive are integrated in support of goal-directed behaviors [2]. The ACC appears to be critically involved in various neurological and psychiatric conditions [3]. Schizophrenia is frequently characterized as a disorder of cognitive and emotional integration [4], and the ACC appears to be central to this pathophysiology. Schizophrenia is emerging as a major public health concern, even in late life [5]. Dysfunction of the ACC has also been reported in geriatric depression [6]. The number of older individuals with major psychiatric disorders is expected to increase due to the aging of the population.

Meanwhile, previous behavioral observations and brain function studies have demonstrated that neurological differences exist between East Asian and Western populations. Neuroimaging studies have revealed cultural differences in neural correlations with cognition and behavior [7,8]. These studies, comparing functional magnetic resonance imaging (MRI) results between East Asian and Western participants, found stable differences in attention, categorization, and contextual processing between the two groups. Thus, ACC analysis focusing on East Asians appears necessary.

Recent neuroimaging studies examining the effects of aging and neuropsychiatric disorders on the cerebral cortex have largely been based on measures of cortical morphometry. The utility of cortical morphometry using measurements of the ACC has been reported for the evaluation of diseases such as schizophrenia and geriatric depression [9,10]. Using measurable parameters of the cerebral cortex such as common surface metrics (cortical thickness, volume, and surface area) as well as parameters of sulcal morphology (cortical folding and curvature), studies have assessed microstructural or functional changes in the regional brain cortex. Meanwhile, neuroimaging studies have largely been based on measures of brain volume. Brain atrophy is a common finding on neuroimaging and is part of the normal aging process, particularly in late middle to advanced age in healthy individuals [11]. On the other hand, accelerated atrophy is associated with cognitive declines and diseases such as dementia [12]. Volumetric studies examining the effects of aging and neurological diseases on brain atrophy are thus important. Brain volume was previously reported to be highly heritable, even in late middle- and advanced-age populations, in a twin study [13].

Twin studies have clearly demonstrated significant genetic contributions to variation in brain structure using measurement of cortical volume, cortical density, or whole-brain/lobar volumes [14]. By examining differences in the degree of similarity between monozygotic (MZ) and dizygotic (DZ) twins, the relative influences of genes (i.e., heritability) and environmental factors on variance in a specific phenotype can be gauged [15]. Further, this method can be used to determine the magnitude of genetic and environmental covariance between phenotypes. In other words, the degree to which phenotypes share common genetic and/or environmental influences can be estimated. Such estimates refer to genetic and environmental correlations, respectively.

We estimated that changes to cortical structures in middle- to advanced-age populations might be more affected by acquired environmental factors as part of the normal aging process than are younger populations. Middle- to advanced-age populations are therefore the focus of this study. By contrast, a previous twin study reported that changes in the parameters of surface morphology such as cortical thickness and surface area are mainly the result of genetic factors [16]. As researchers continue to examine neurological and psychiatric conditions influencing brain structure in middle to advanced age, recognizing that a selected parameter may be influenced by multiple underlying factors such as genetics or aging is critical. To utilize surface morphological parameters of the ACC as an imaging biomarker for evaluating diseases in middle- to advanced-age populations, we considered that the contribution of each genetic and environmental factor to these parameters should be analyzed to clarify the basic characteristics of such parameters. The aim of the present study was therefore to clarify the characteristics of surface morphometric parameters for quantifying ACC changes in middle- to advanced-age populations using an East Asian twin cohort.

## 2. Materials and Methods

This study was approved by the ethics committee of our institution, and written informed consent was obtained from all subjects after they received an explanation of the purposes of the study and of the possible consequences of participating in the study. 

### 2.1. Subjects

Our twin registry comprised a cumulative total of over 500 twin pairs participating in the core survey. All the subjects volunteered to participate in the study. Zygosity was confirmed using 15 loci of short tandem repeat (STR) markers, with complete concordance of these STRs diagnostic of a monozygotic twin pair. MRI data were obtained from each subject between January 2011 and December 2014 as a part of a neuroscientific study based on our aged-twin registry. We targeted asymptomatic normal pairs of twins, and 45 monozygotic and 18 dizygotic twins underwent brain MRIs. As a subset for this study of ACC analysis in a middle- to advanced-age normal population, subjects under 40 years old were excluded. According to the inclusion and exclusion criteria (Figure 1), a total of 108 participants were selected from the database and were age matched, comprising 37 monozygotic twin pairs and 17 dizygotic twin pairs. The monozygotic twin pairs comprised 10 male–male pairs and 27 female–female pairs, and the dizygotic twin pairs comprised 8 male–male pairs, 8 female–female pairs and 1 male–female pair (Table 1). The median age at the time of the MRI was 61 years (range, 42–75 years) for the monozygotic twin pairs and 67 years (range, 41–83 years) for the dizygotic twin pairs, respectively. The Mini-Mental State Examination was used to assess cognitive function, with an overall mean score for the participants of 28.1 (range, 22–30). 

### 2.2. Imaging Studies

Three-dimensional (3D) T1-weighted imaging of the brain was performed at a single center but used two types of MR scanner, due to equipment changes. Thirty-four monozygotic twin pairs and 6 dizygotic twin pairs were scanned on a 3.0-T Signa HDxt scanner (GE Healthcare, Milwaukee, WI, USA), and sagittal images were obtained using the following scan conditions: echo time (TE), 2.9 ms; repetition time (TR), 7.0 ms; inversion time (TI), 400 ms; in-plane resolution, 1 × 1 mm; slice thickness, 1 mm; slice number, 180; and flip angle, 11°. Three monozygotic twin pairs and 11 dizygotic twin pairs were scanned on a 3.0-T Achieva scanner (Philips Healthcare, Best, The Netherlands), and sagittal images were obtained using the following conditions: TE, 3.1 ms; TR, 6.7 ms; TI, 880 ms (shortest time in settings); in-plane resolution, 1 × 1 mm; slice thickness, 1 mm; slice number, 180; and flip angle, 8°. All the individuals from a specific twin pair were scanned using the same MRI scanner on the same day. Two neuroradiologists with, respectively, 20 and 30 years of experience in radiology checked all the brain images to confirm the absence of structural abnormalities.

### 2.3. Image Analysis

We used a surface-based protocol for parcellating the ACC. In a preprocessing step, 3D T1-weighted volume images were exported as DICOM files, then converted to FSL 4D NIfTI files. Automated cortical surface reconstruction and estimation of cortical morphological parameters for the ACC were performed on preprocessed 3D T1-weighted volume image data using the FreeSurfer version 7.1.1 image analysis suite (http://surfer.nmr.mgh.harvard.edu: Charlestown, MA, USA.; Accessed on 11 November 2020) [17]. The ACC region was defined using Destrieux Atlas.

FreeSurfer-derived parameters including common surface metrics (cortical thickness, STDthickness, and volume and surface area), as well as parameters of sulcal morphology (cortical folding, mean curvature, and Gaussian curvature) were applied for the ACC analysis. Thickness was defined as the averaged distance between linked vertices on the white matter and pial surface in each brain region. STDthickness was defined as the variation in thickness in each brain region. Volume was defined as the space between the white matter and the pial surface. FreeSurfer represented the smoothed brain surface as a triangle-based mesh, so surface area was defined as the sum of areas of the triangles in each brain region. Folding in each brain region was calculated as the ratio of the cortical area contained in a sphere divided by the area of a disc of the same radius. The smoothed brain surface was inflated to produce the inflated surface. Subsequently, the unit normal vector at each vertex *v* of the inflated surface was calculated. A normal plane at vertex *v* contains a unique direction tangent to the inflated surface and cuts the surface in a plane curve. This curve would generally have different curvatures for different normal planes at vertex *v*. The principal curvatures at *v*, denoted *k1* and *k2*, were the maximum and minimum values of this curvature. Based on these values, mean curvature and Gaussian curvature were then defined from the following equations: mean curvature = 1/2(*k1* + *k2*) and Gaussian curvature = *k1* × *k2*.

### 2.4. Data Analysis

Both statistical and twin analyses were performed for each parameter of the bilateral ACCs.

#### 2.4.1. Statistical Analysis

Correlations with age were assessed by calculating the Pearson correlation coefficient, and differences based on each zygosity and laterality between left and right ACCs in all the subjects were also assessed with the Wilcoxon signed-rank test using statistical package for the social sciences software (IBM SPSS Statistics for Windows, version 27.0; IBM, Armonk, NY, USA). A two-sided *p*-value less than 0.05 was considered to indicate statistical significance.

#### 2.4.2. Twin Analysis

The ACE model is a statistical model commonly used for twin analysis, decomposing sources of phenotypic variation into three categories: additive genetic variance (A); shared environmental factors common to both members of a twin pair (C); and non-shared or unique environmental factors plus measurement error (E) [18]. This model is widely used in genetic epidemiology and behavioral genetics. 

As an initial step, intra-twin pair correlations were calculated for the intraclass correlation coefficients (ICCs), also assessed by calculating the Pearson correlation coefficient, and compared for monozygotic and dizygotic twin groups using the R version 4.1.0 statistical platform (https://www.r-project.org/: The University of Auckland, Auckland, New Zealand; Accessed on 11 November 2020). In addition, to assess the effect of aging on the inter-zygosity difference in the ICCs, the significance of the difference in age between zygosities was tested using a two-tailed Mann–Whitney U test.

Next, intra-twin differences within monozygotic and dizygotic twin pairs were compared to calculate the contribution of each genetic and environmental factor to the phenotype with a twin analysis model. Classical twin studies assume that monozygotic twins share 100% and dizygotic twins share 50% of segregating genes [19]. According to this underlying assumption, structural equation models were constructed using OpenMx version 3.9.2 (Virginia Institute for Psychiatric and Behavior Genetics, Virginia Commonwealth University, Richmond, VA) in R. The three phenotype variance components of additive genetic effects (A), common environmental effects (C), and unique environmental effects (E) were used to produce four models: ACE, AE, CE, and E. These models were then compared for each parameter assessment based on fit and simplicity using the Akaike information criterion (AIC) [20]. The model showing the lowest AIC was selected as the model of best fit. In addition, to control for variations in age and sex, these factors were included in structural equation models as covariates.

## 3. Results 

### 3.1. Statistical Analysis

Table 2 lists the Pearson correlation coefficients of each parameter with age. Bilateral STDthickness and curvatures and right folding demonstrated significant positive correlations with age, whereas bilateral thickness, volume, surface area, and left folding did not. (The Appendix A show the correlations between parameter value and age in the left and right ACC, respectively).

Table 3 and Table 4 list the differences in the parameters based on zygosity and laterality, respectively. No significant differences in zygosity were apparent in any parameters. For laterality, left thickness was significantly higher than right thickness. Left STDthickness was significantly higher than right STDthickness. The left volume was significantly lower than the right volume. The left surface area was significantly lower than the right surface area. The left folding was significantly lower than the right folding.

### 3.2. Twin analysis

Table 5 lists the ICCs of the parameters for each zygosity. The higher ICC for monozygotic twins compared with dizygotic twins for all the values of the ACC parameters confirmed the validity of our data for performing twin analysis. No significant inter-zygosity difference was seen in age, indicating no significant effect of aging on the ICC. Figure 2 and Figure 3 show intra-twin pair correlations for the left ACC parameters. (In addition, the Appendix A show intra-twin pair correlations for the left and right ACC parameters).

Table 6 lists the ACE analysis estimate for each A, C, and E factor in each model of all the parameters. Based on the AIC, most parameters fit best with an AE model, supporting significant contributions from genetic factors. In contrast, right volume and left Gaussian curvature fit best with a CE model, indicating relatively weak contributions of genetic factors to these parameters. 

Thickness demonstrated high genetic contributions of 0.73 in the left and 0.71 in the right. Surface area demonstrated high genetic contributions of 0.63 in the left and 0.71 in the right. By contrast, STDthickness demonstrated low to moderate genetic contributions of 0.48 in the left and 0.29 in the right. Left volume demonstrated a moderate genetic contribution of 0.37.

Folding demonstrated low to moderate genetic contributions of 0.44 in the left and 0.28 in the right. Right mean curvature and right Gaussian curvature demonstrated high heritable contributions of 0.65 and 0.79, respectively. Conversely, left mean curvature demonstrated moderate genetic contribution of 0.37.

## 4. Discussion 

To date, multiple twin studies have demonstrated highly regionalized expression patterns of genes in the entire brain cortical structure. However, few studies have investigated characteristics of specific regional cortical structures, such as the ACC of East Asian populations.

Structural abnormalities of the ACC have been demonstrated in neuropsychiatric disorders such as schizophrenia. Some studies have shown volume reductions in the bilateral ACCs of patients with schizophrenia [21,22]. Fujiwara et al. observed multiple structural abnormalities of the ACC in schizophrenia, including changes in volume, white matter microstructures, and macroscopic sulcal morphology. They also suggested that different dimensions of psychopathology and social cognitive function are associated in different ways with cortical and sulcal morphological abnormalities in the ACC [23], suggesting the importance of investigating differences in the characteristics of surface morphology.

Functional decreases in the ACC were observed as large declines in brain activity with normal aging according to measurements with fluorodeoxyglucose and positron emission tomography [24], indicating the importance of considering the effects of aging on the ACC. Age-related changes in both cortical thickness and surface area are well established [25]. By contrast, the present study revealed statistically nonsignificant tendencies toward slight negative correlations with age in thickness, volume, and surface area in bilateral ACCs. We attributed this to the limitation on age range to middle to advanced age. STDthickness, indicating variation in thickness, showed a positive correlation with normal aging, aligning with the results of twin analysis that the environmental contribution to STDthickness was relatively strong in middle to advanced age. Parameters of sulcal morphology such as folding and curvatures were the most sensitive to age-related ACC changes. 

While research on hemispheric asymmetry can establish functional lateralization of the brain, the nature of structural and functional differences between hemispheres remains poorly understood. The present study showed significant asymmetry in each of surface area, thickness, volume, STDthickness, and folding. A previous study reported similar tendencies in the cingulate gyrus, with asymmetry in surface area, thickness, and volume in young adults [26]. 

In addition, the present twin analysis revealed that both the cortical thickness and the surface area of the ACC were highly heritable. Previous studies of the entire brain structure have also reported both cortical thickness and surface area as highly heritable [27]. These findings indicate that the underlying genetic architecture of the ACC should be considered when evaluating neurological diseases in genetically informative studies. According to previous reports, cortical thickness and surface area are two constituent components of cortical structure that are considered genetically distinguishable features of brain morphology. Cortical thickness and surface area have been utilized for evaluating Williams syndrome, a specific genetic disorder, by quantifying brain regions including the anterior cingulum [28]. Schizophrenia is a neurodevelopmental disorder with high heritability, up to 80% [29]. A previous study indicated the usefulness of both cortical thickness and surface area for detecting ACC abnormalities in schizophrenia [30]. Accordingly, both cortical thickness and surface area might be clinically useful for evaluating genetic disorders or diseases with suspected strong genetic effects mainly in younger generations. In addition, the present study revealed that both cortical thickness and surface area are strongly affected by genetic factors even in middle- to advanced-age populations, which may indicate the potential utility of these two parameters in evaluating such diseases in later life.

By contrast, genetic effects on ACC volume were not strong in our results. The genetic contribution was moderate for the left ACC, and the CE model was applied for the right ACC. By definition, brain volume represents the product of cortical thickness and surface area. Cortical volumetry could thus combine structural properties that are unique to cortical thickness and unique to surface area. The present twin analysis revealed that both the cortical thickness and the surface area of the ACC are highly heritable. Meanwhile, according to a previous twin analysis, distinct genetic influences on surface area and cortical thickness exist in the entire brain region [31], which may confound the underlying genetic effect on ACC volume. Cortical volumetry has been widely utilized in previous studies. Reductions in the cortical volume of the ACC have been reported in patients with schizophrenia [21,22], although these results remain controversial [32,33,34]. One study reported the clinical application of measuring changes in ACC volume for assessing the outcomes of treatment for geriatric depression [10]. Although the biological contributions to geriatric depression remain unclear, both normal aging and aging-related brain changes are associated with the development of late-life depression [6]. Cortical volumetry may be more useful for evaluating changes in normal aging or age-related diseases compared to changes in genetic disorders.

In terms of sulcal morphology, folding and curvatures mostly showed statistically significant relationships with normal aging and low to moderate genetic contributions in this middle- to advanced-age population. Age-related trends in the sulcal morphology of the ACC were shown as increased folding values with increased mean and Gaussian curvature values, consistent with the age-related trends reported from a previous study [35]. A previous twin study examining the heritability of cortical folding revealed low heritability for the ACC on a heritability brain map [36]. Another study identified no significant changes for regional grey matter volume or sulcal morphologies such as curvature in patients with schizophrenia [30]. Whether parameters of sulcal morphology are useful for evaluating normal aging or age-related diseases thus remains unclear. Sulcal morphologies such as gyral curvature have been measured as neuroimaging markers of age-related brain atrophy [37]. Age is one of the main risk factors for dementia. Compared with other cortical parameters such as thickness, one study found that parameters of sulcal morphology such as mean curvature offered the highest sensitivity for detecting early dementia [38]. Accordingly, we consider the parameter of sulcal morphology mainly to measure age-related changes in the ACC, meaning that the ACC sulcal morphological parameter may be useful for evaluating normal aging or age-related diseases.

This study showed several limitations that might have affected the results. First, the sample size was small. Further studies with an increased number of subjects, especially for dizygotic twins, may be beneficial. Next, equipment changes in the department resulted in the twin group undergoing a brain MRI using two different 3-T platforms, which may have caused imaging variations that would have introduced biases to subsequent cortical measurements based on the imager. The subjects were selected to satisfy age requirements, but the age range was relatively wide and brain heritability may have varied within this age range. In addition, inter-sex differences were not assessed, because numbers were not matched between sexes in the monozygotic twin group. Acknowledging these problems, each twin pair underwent an MRI examination using the same platform on the same day, which is important for twin analyses examining intra-twin differences. Age and sex were included as covariates in the structural equation models used to produce the final heritability estimates.

## 5. Conclusions

In middle- to advanced-age East Asian populations, surface morphological parameters for quantifying changes in the ACC appear to be affected by genetic factors, and environmental factors such as chronic adversity. The magnitude of influence for each factor depends on the specific parameter. In particular, both cortical thickness and surface area are strongly affected by genetic factors, while the parameter of sulcal morphology tends to relate to environmental factors with aging.

## Figures and Tables

**Figure 1 medicina-58-01855-f001:**
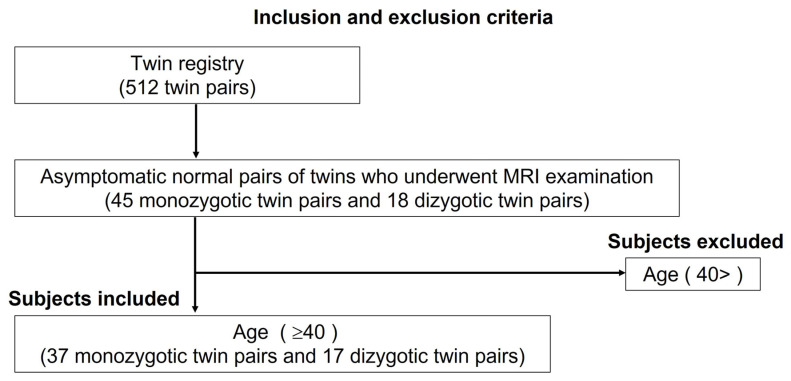
Inclusion and exclusion criteria for subjects.

**Figure 2 medicina-58-01855-f002:**
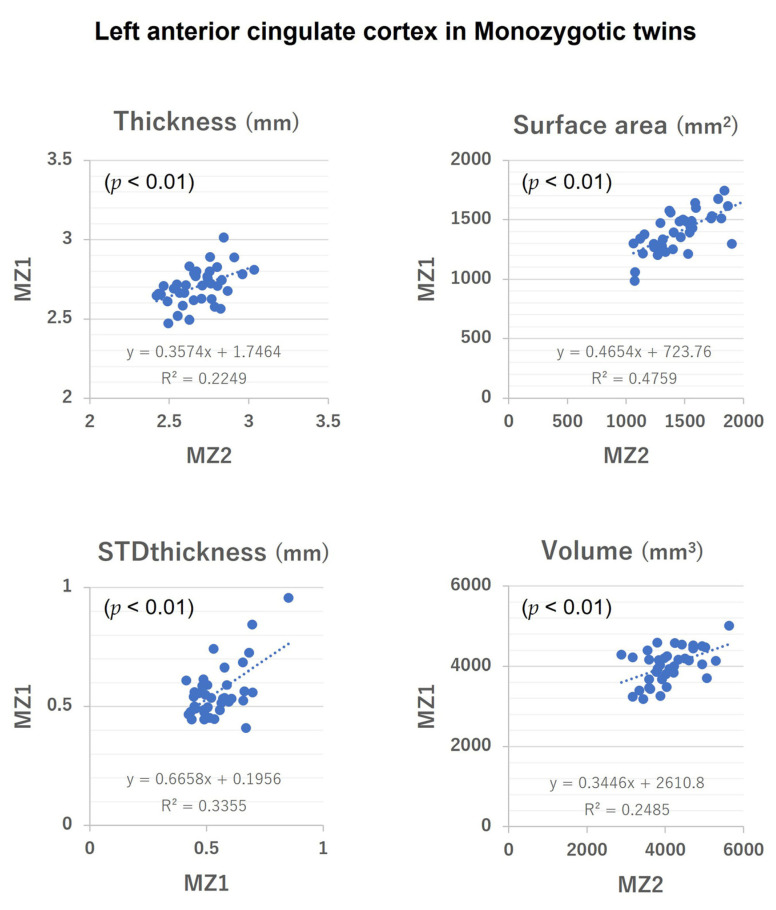
Intra-twin pair correlation of surface morphological parameter values (Thickness, Surface area, STDthickness and Volume) of the left anterior cingulate cortex in monozygotic twins. STDthickness: standard deviation of the thickness.

**Figure 3 medicina-58-01855-f003:**
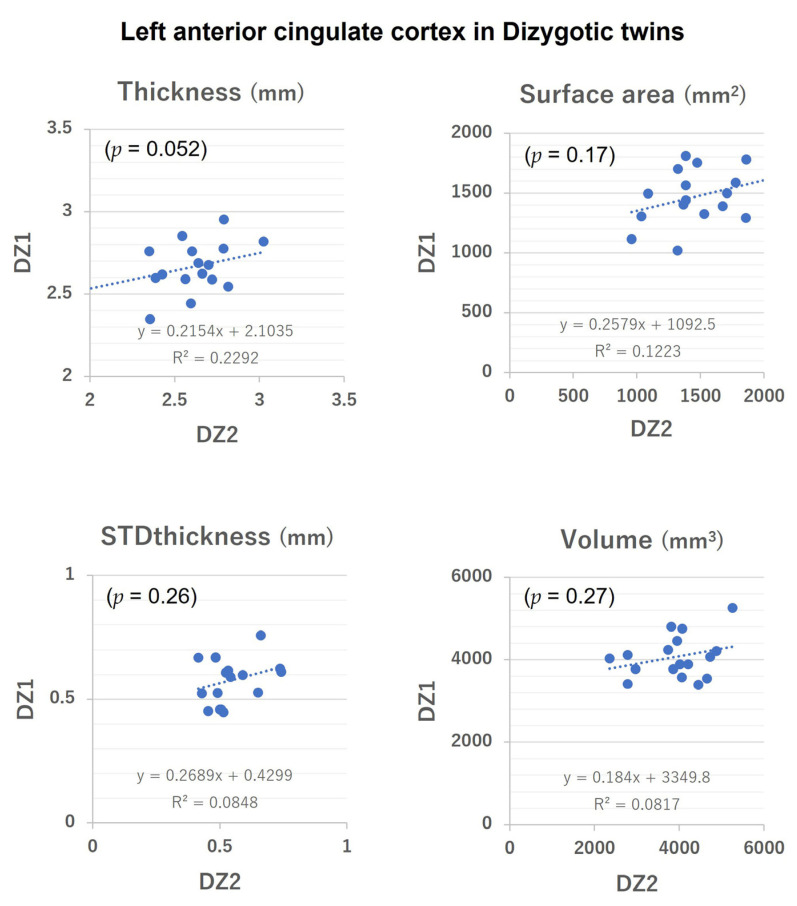
Intra-twin pair correlation of surface morphological parameter values (Thickness, Surface area, STDthickness, and Volume) of the left anterior cingulate cortex in dizygotic twins. STDthickness: standard deviation of the thickness.

**Table 1 medicina-58-01855-t001:** Subjects.

	Monozygotic Twin	Dizygotic Twin
Twin pair: Pair (Number)	37 (74)	17 (34)
Gender: Male/Female	20/54	17/17
Age: Median (Range) years	61 (42–75)	67 (41–83)

**Table 2 medicina-58-01855-t002:** Correlation of surface morphological parameters with age.

	Left ACC		Right ACC	
	R Value	*p*-Value	R Value	*p*-Value
Thickness	−0.09	0.35	−0.13	0.17
STDthickness	0.37 *	<0.05	0.29 *	<0.05
Volume	−0.11	0.26	−0.06	0.52
Surface area	−0.03	0.74	0.05	0.64
Folding	0.13	0.15	0.31 *	<0.05
Meancurv	0.31 *	<0.05	0.33 *	<0.05
Gauscurv	0.45 *	<0.05	0.36 *	<0.05

ACC: anterior cingulate cortex, STDthickness: standard deviation of the thickness, Meancurv: mean curvature, Gauscurv: gaussian curvature. *: Significant correlation with age (*p* < 0.05) and *p*-values.

**Table 3 medicina-58-01855-t003:** Differences in surface morphological parameters in zygosity.

Left ACC				
	MZ		DZ	
	Mean	SD	Mean	SD
Thickness (mm)	2.69	0.13	2.60	0.27
STDthickness (mm)	0.55	0.11	0.56	0.09
Volume (mm^3^)	4065.05	542.02	3992.71	669.15
Surface area (mm^2^)	1430.47	215.74	1476.91	262.80
Folding	23.19	5.19	22.88	5.81
Meancurv (mm^−1^)	0.13	0.01	0.12	0.01
Gauscurv (mm^−2^)	0.03	0.01	0.03	0.01
**Right ACC**				
	**MZ**		**DZ**	
	**Mean**	**SD**	**Mean**	**SD**
Thickness (mm)	2.65	0.12	2.58	0.25
STDthickness (mm)	0.50	0.06	0.52	0.06
Volume (mm^3^)	5227.39	591.16	5151.76	635.09
Surface area (mm^2^)	1897.09	228.40	1937.26.	237.89
Folding	30.89	5.40	30.79	6.73
Meancurv (mm^−1^)	0.13	0.01	0.13	0.01
Gauscurv (mm^−2^)	0.03	0.01	0.03	0.01

ACC: anterior cingulate cortex, MZ: mono zygosity, DZ: dizygosity, STDthickness: standard deviation of the thickness, Meancurv: mean curvature, Gauscurv: gaussian curvature. SD: standard deviation.

**Table 4 medicina-58-01855-t004:** Differences of surface morphological parameters in laterality.

	Left ACC		Right ACC		
	Mean	SD	Mean	SD	*p*-Value
Thickness (mm)	2.66 *	0.19	2.63 *	0.17	<0.05
STDthickness (mm)	0.56 *	0.1	0.51 *	0.06	<0.05
Volume (mm^3^)	4042.28 *	582.81	5203.58 *	603.38	<0.05
Surface area	1445.09 *	231.35	1909.74 *	243.06	<0.05
Folding	23.09 *	5.36	30.86 *	5.82	<0.05
Meancurv (mm^−1^)	0.12	0.01	0.13	0.01	0.06
Gauscurv (mm^−2^)	0.03	0.01	0.03	0.01	0.73

ACC: anterior cingulate cortex, STDthickness: standard deviation of the thickness, Meancurv: mean curvature, Gauscurv: gaussian curvature. SD: standard deviation. *: Significance of difference (*p* < 0.05) and *p*-values.

**Table 5 medicina-58-01855-t005:** Intraclass correlation coefficients of surface morphological parameters.

	Left ACC		Right ACC	
	MZ (95% CI)	DZ (95% CI)	MZ (95% CI)	DZ (95% CI)
Thickness (mm)	0.46 (0.21–0.65)	0.35 (−0.037–0.65)	0.4 (0.14–0.60)	0.34 (−0.067–0.65)
STDthickness (mm)	0.58 (0.36–0.73)	0.29 (−0.11–0.61)	0.37 (0.12–0.58)	0.054 (−0.34–0.44)
Volume (mm^3^)	0.47 (0.23–0.66)	0.27 (−0.14–0.60)	0.59 (0.38–0.74)	0.41 (0.048–0.68)
Surface area	0.62 (0.43–0.77)	0.36 (−0.039–0.66)	0.68 (0.51–0.89)	0.40 (0.011–0.69)
Folding	0.43 (0.18–0.62)	0.41 (0.024–0.70)	0.34 (0.080–0.56)	0.27 (−0.11–0.59)
Meancurv (mm^−1^)	0.45 (0.20–0.64)	0.34 (−0.050–0.65)	0.62 (0.42–0.76)	0.20 (−0.21–0.55)
Gauscurv (mm^−2^)	0.66 (0.47–0.79)	0.57 (0.24–0.79)	0.80 (0.67–0.88)	0.67 (0.37–0.84)

ACC: anterior cingulate cortex, CI: Confidence Interval, STDthickness: standard deviation of the thickness, Meancurv: mean curvature, Gauscurv: gaussian curvature.

**Table 6 medicina-58-01855-t006:** Structural equation modelling results of surface morphological parameters.

		Left ACC				Right ACC			
	Model	AIC	A	C	E	AIC	A	C	E
**Thickness:**	ACE	573.14	0.43	0.57	0	−77.19	0.71	0	0.29
	AE	**−59.01**	**0.73**	**0**	**0.27**	**−79.79**	**0.71**	**0**	**0.29**
	CE	−52.25	0	0.44	0.56	−73.28	0	0.39	0.61
	E	−43.38	0	0	1	1690.8	0	0	1
**STDthickness:**	ACE	−202.93	0.48	0	0.52	−304.13	0.29	0	0.71
	AE	**−204.94**	**0.48**	**0**	**0.52**	**−306.13**	**0.29**	**0**	**0.71**
	CE	−203.18	0	0.42	0.58	−304.85	0	0.2	0.8
	E	−194.89	0	0	1	−304.56	0	0	1
**Volume:**	ACE	1666.48	0.29	0.07	0.64	1678.58	0.57	0.01	0.43
	AE	**1664.42**	**0.37**	**0**	**0.63**	1676.6	0.57	0	0.43
	CE	1665.12	0	0.28	0.72	**1676.34**	**0**	**0.52**	**0.48**
	E	1667.365	0	0	1	1690.8	0	0	1
**Surface area:**	ACE	1466.44	0.63	0	0.37	1479.38	0.71	0	0.29
	AE	**1464.44**	**0.63**	**0**	**0.37**	**1477.38**	**0.71**	**0**	**0.29**
	CE	1469.00	0	0.43	0.57	1483.07	0.54	0.46	0
	E	1477.06	0	0	1	1498.61	0	0	1
**Folding:**	ACE	665.42	0.32	0.11	0.57	684.54	0.28	0	0.72
	AE	**663.49**	**0.44**	**0**	**0.56**	**682.54**	**0.28**	**0**	**0.72**
	CE	663.89	0	0.37	0.63	682.87	0	0.21	0.79
	E	670.02	0	0	1	683.2	0	0	1
**Meancurv:**	ACE	−675.27	0.27	0.1	0.63	−693.53	0.65	0	0.35
	AE	**−677.25**	**0.37**	**0**	**0.63**	**−693.53**	**0.65**	**0**	**0.35**
	CE	−677.09	0	0.34	0.66	−688.65	0	0.36	0.64
	E	−672.4	0	0	1	−683.35	0	0	1
**Gauscurv:**	ACE	−802.12	0.24	0.31	0.44	−843.17	0.44	0.35	0.22
	AE	−803.67	0.57	0	0.43	**−843.95**	**0.79**	**0**	**0.21**
	CE	**−803.77**	**0**	**0.53**	**0.47**	−841.88	0	0.7	0.3
	E	−788.39	0	0	1	−592.74	0	0	1

Values of chosen models based on the lowest AIC values were depicted in bold letters. ACC: anterior cingulate cortex, STDthickness: standard deviation of the thickness, Meancurv: mean curvature, Gauscurv: gaussian curvature, A: additive genetic effects, C: common environmental effects, E: unique Environmental effects, AIC: Akaike information criterion.

## Data Availability

The dataset used for the study is available from the corresponding author on reasonable request.

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
