# Peer review of "Assessment of Characteristics of Imaging Biomarkers for Quantifying Anterior Cingulate Cortex Changes: A Twin Study of Middle- to Advanced-Aged Populations in East Asia"

_medicina, 2022, doi:10.3390/medicina58121855_

Round 1

Reviewer 1 Report

1. Are changes in ACC ethnically related? Are there other ethnically relevant studies?

2. The authors should highlight the limitation in the discussion section.

3. Quality of the figures and tables must be improved.

4. The manuscript needs English proofreading.

Author Response

  1. Are changes in ACC ethnically related? Are there other ethnically relevant studies?

*Thank you for your question. According to the paper of reference No.7 in the manuscript (Joan Y.C. et al. Cultural Neuroscience: Progress and Promise. Psychol Inq 2013, 24, 1-19.), racial identification is associated with increased neural response within cortical midline structures including anterior cingulate cortex (ACC) and posterior cingulate cortex (PCC). Other study reported that culture and genes regulate psychological and neural mechanisms of mental state understanding. For interdependent cultures, other-focusedness is associated with greater neural response within empathic neural circuitry, such as the anterior cingulate cortex (ACC) and insula (Cheon BK, et al.Neuropsychologia, 2013, 51(7), 1177–86.). Therefore, we consider that there is the ethnically related difference including the ACC function.

  1. The authors should highlight the limitation in the discussion section.

*According to your suggestion, instead of the current phrase, we highlight the limitation with the phrase that this study showed several limitations that might affect the results.

  1. Quality of the figures and tables must be improved.

*Thank you for your suggestion, we are going to improve the quality of figures.

  1. The manuscript needs English proofreading.

*Thank you for your suggestion, we are going to take the manuscript check by the native English speaker.

Reviewer 2 Report

 Hiroto Takahashi and colleagues presented an interesting study on ACC morphology. The paper is well-written, and the results are plausible. However, before accepting it, I would suggest three minor improvements.

1.       Authors should specify in the methods what are environmental and unique environmental factors.

2.       The small-vessel disease also affects ACC volume. Maybe I missed a report on how many subjects had small-vessel disease, e.g. table with Fazekas 0 to 3.

3.       In conclusion, the authors should specify environmental factors. This information is relevant for clinical practice.

Author Response

  1. Authors should specify in the methods what are environmental and unique environmental factors.

*Thank you for your suggestion. We modified the explanation for those.

  1. The small-vessel disease also affects ACC volume. Maybe I missed a report on how many subjects had small-vessel disease, e.g. table with Fazekas 0 to 3.

*Thank you for your suggestion. When performing the image analysis, we have checked the brain T2w images to exclude any brain abnormalities. And there were no significant findings including white matter lesions (suspicious demyelination or vessel diseases).

  1. In conclusion, the authors should specify environmental factors. This information is relevant for clinical practice.

*According to your suggestion, we added the information regarding the acquired environmental factors (Int J Geriatr Psychiatry 2008, 23, 347-355.).

Reviewer 3 Report

The authors assessed genetic and environmental effects on surface morphological parameters for quantifying anterior cingulate cortex (ACC) changes in middle- to advanced-aged East Asian (37 monozygotic pairs and 17 dizygotic pairs) using twin analysis. They investigated the freesurfer-derived ACC parameters calculated from 3D T1- weighted volume images including thickness, standard deviation of thickness, volume, surface area and sulcal morphological parameters. They used ACE model to assess the magnitude of each genetic and environmental influence on the ACC parameters. The authors concluded that ACC thickness and surface area appear strongly affected by genetic factors, whereas sulcal morphological parameters tend to involve environmental factors. The paper has the potential to contribute to the existing scientific literature on ACC changes in middle- to advanced-aged East Asian. I only have a few comments to further improve the quality of the authors’ paper. I have outlined these issues below:

1.

Most of the tables should be reorganized. The data in the tables are not well lined up.

2.

Figure 2 and 3,  no p value was provided for Intra-twin pair correlation

3.

Page 5, line

The ACE model is a statistical model that is commonly used for twin analysis, 180 decomposing sources of phenotypic variation into three categories: additive genetic variance (A), common environmental factors (C), and unique environmental factors plus measurement error (E) [18], and is widely used in genetic epidemiology and behavioural genetics.

The readers may wonder what are the factors for C and E?

4.

Page 12, line 399

A previous study indicated the usefulness of both cortical thickness and surface area for detecting ACC abnormalities in schizophrenia [30].

The reference is dated and can be updated.

5.

In the method,

Three-dimensional (3D) T1-weighted imaging of the brain was performed at a single 129 center but used two types of MR scanner, due to equipment changes.

In the discussion

Next, equipment changes in the department resulted in the twin group undergoing brain MRI using two different 3-T platforms, which may have caused imaging variations that would have introduced biases to subsequent cortical measurements based on the imager.

Is it possible for the author to use different equipment as a variable and manage this potential bias with a statistical method. If not, the limitation mentioned in the discussion is acceptable.

6.

Instead of defining the volume and thickness of the cortex between the gray- and white-matter surfaces, it would be more appropriate to define the thickness and volume of the cortex between the two.

7. Have you corrected the effect of multiple comparisons on the statistical analysis? If no, it is recommended that you check the results of the statistics with multiple comparison corrections.

8. Have you corrected the effect of multiple comparisons on the statistical analysis? If no, it is recommended that you check the results of the statistics with multiple comparison corrections.

In the reviewer’s opinion, the above-mentioned issues need to be addressed by the authors.

Author Response

  1. Most of the tables should be reorganized. The data in the tables are not well lined up.

*According to your suggestion, we modified the table.

  1. Figure 2 and 3,  no p value was provided for Intra-twin pair correlation

*According to your suggestion, we added the p-value for each.

3.Page 5, line

The ACE model is a statistical model that is commonly used for twin analysis, 180 decomposing sources of phenotypic variation into three categories: additive genetic variance (A), common environmental factors (C), and unique environmental factors plus measurement error (E) [18], and is widely used in genetic epidemiology and behavioural genetics.

The readers may wonder what are the factors for C and E?

*Thank you for your suggestion. We modified the explanation for those.

4.Page 12, line 399

A previous study indicated the usefulness of both cortical thickness and surface area for detecting ACC abnormalities in schizophrenia [30]. The reference is dated and can be updated.

*Thank you for your suggestion. We checked the reference again and updated this.

5.In the method, Three-dimensional (3D) T1-weighted imaging of the brain was performed at a single 129 center but used two types of MR scanner, due to equipment changes.

 In the discussion

Next, equipment changes in the department resulted in the twin group undergoing brain MRI using two different 3-T platforms, which may have caused imaging variations that would have introduced biases to subsequent cortical measurements based on the imager.

Is it possible for the author to use different equipment as a variable and manage this potential bias with a statistical method. If not, the limitation mentioned in the discussion is acceptable.

*Thank you for your suggestion. We tried to evaluate the difference of the equipment as a covariate in the statistical analysis including the ACE analysis. However, the detail of the difference could not be defined. So, we described the difference to the limitation section. The difference might affect the measurement of the cortex. In contrast, we assessed the images visually, and apparent difference in the image quality was not observed. Therefore, we considered that the difference of the equipment was small effect on the measurement of the cortex.

  1. Instead of defining the volume and thickness of the cortex between the gray- and white-matter surfaces, it would be more appropriate to define the thickness and volume of the cortex between the two.

*We agree with your opinion.

  1. Have you corrected the effect of multiple comparisons on the statistical analysis? If no, it is recommended that you check the results of the statistics with multiple comparison corrections.
  2. Have you corrected the effect of multiple comparisons on the statistical analysis? If no, it is recommended that you check the results of the statistics with multiple comparison corrections.

 *Thank you for your suggestion. Intergroup difference for each value was assessed with the Wilcoxon signed-rank test. We did not consider the effect of multiple comparisons, because of intergroup difference between the two groups (MZ vs DZ and each value vs age or sex. etc.).

Round 2

Reviewer 3 Report

 The authors address all the reviewer's comments carefully and have improved their manuscript considerably. I recommend this manuscript for publication.